# GSRQ: Gain–Shape Residual Quantization for Sub-1-bit KV Cache

**Soosung Kim** [* 1]   **Minjae Park** [* 2]   **Eui-Young Chung** [1]   **Jaeyong Chung** [2 1]

## Abstract

The deployment of Large Language Models (LLMs) with extended context windows is increasingly constrained by the linear growth of Key–Value (KV) cache memory. Vector Quantization (VQ), particularly Residual Quantization (RQ), is a promising approach for pushing KV cache storage toward the sub-1-bit regime by progressively encoding residuals with small codebooks. However, most VQ methods still rely on standard $\ell_2$ $K$-means as the core codebook-learning primitive. We identify a subtle high-dimensional issue of this primitive: Euclidean centroid averaging can induce centroid shrinkage, which weakens the angular alignment term in the $\ell_2$ distortion and makes directional preservation harder. To address this issue, we propose Gain–Shape $K$-means (GSKM), a drop-in replacement for $K$-means that improves directional fidelity while matching, and in some regimes improving, $\ell_2$ distortion. We then build Gain–Shape Residual Quantization (GSRQ) by incorporating a weighted extension of GSKM into an RQ pipeline. On LLaMA-3-8B, GSRQ substantially improves over strong KV cache quantization baselines across bit rates. At 1-bit, it improves the average accuracy across LongBench tasks from 11.34 to 33.54, a gain of 22.20 percentage points over VQLLM.

## 1. Introduction

Large Language Models (LLMs) have demonstrated strong performance across various tasks, including complex reasoning and code generation. Extending the context window of these models is essential for processing long documents, large-scale codebases, and complex dependencies. However,

long-context inference introduces a significant memory bottleneck, particularly in deployment scenarios where high throughput is required through batch processing. This overhead is primarily caused by the KV cache, which stores attention states to avoid redundant computation. Since the KV cache size scales linearly with both sequence length and batch size, it often exceeds the available memory of modern GPUs. For instance, LLaMA-3.1-8B fits comfortably in a single H100-80GB GPU with FP16 weights (16GB), but with a 128K context and batch size 8, the KV cache can grow to 128GB and exceed GPU memory. This KV cache growth becomes the primary bottleneck, limiting throughput and practical long-context serving.

To mitigate these demands, post-training compression methods typically employ token eviction/pruning or quantization. Token eviction/pruning reduces the number of tokens to store by retaining only a subset of past tokens, typically guided by a token saliency signal derived from attention statistics (Zhang et al., 2023; Liu et al., 2023; Xiao et al., 2024b). Quantization instead reduces the per-element storage cost by lowering the bit-width (Liu et al., 2024; Hooper et al., 2024; Su et al., 2025). Importantly, saliency can also be used to enable mixed precision, where a small set of salient (or outlier) tokens is stored at higher precision while the remainder is more aggressively quantized (He et al., 2024). Offloading provides an orthogonal system-level knob by moving (parts of) the cache to host memory or disk, but often trades memory for data-movement latency (Sheng et al., 2023; Aminabadi et al., 2022).

Most existing KV cache quantization methods adopt scalar quantization, treating each key/value element independently (Liu et al., 2024; Hooper et al., 2024); for instance, KIVI quantizes the Key cache per-channel and the Value cache per-token to minimize outlier impact. Quantization theory implies that jointly quantizing vectors can provide a higher effective signal-to-noise ratio over scalar quantization even when the source components are i.i.d. Gaussian (Gray & Neuhoff, 2002), motivating vector-quantized KV caches. Vector Quantization (VQ) has a long history in audio/speech coding and large-scale similarity search (Makhoul et al., 1985; Gersho & Gray, 1992). In VQ, a vector is encoded by the index of a nearest codeword from a learned codebook; Product Quantization (PQ) makes this scalable by partitioning the vector into low-dimensional subspaces and

---

[*]Equal contribution   [1]Department of Electrical and Electronic Engineering, Yonsei University, Seoul, Republic of Korea [2]Department of Systems Semiconductor Engineering, Yonsei University, Seoul, Republic of Korea. Correspondence to: Jaeyong Chung <jyc@yonsei.ac.kr>.

*Proceedings of the $43^{rd}$ International Conference on Machine Learning*, Seoul, South Korea. PMLR 306, 2026. Copyright 2026 by the author(s).

quantizing each subvector with its own codebook (Jégou et al., 2011).

Beyond PQ, Additive Quantization (AQ) represents a vector as a sum of multiple codewords drawn from multiple (typically full-dimensional) codebooks, and Residual Quantization (RQ) is its sequential specialization that greedily encodes the residual at each stage (Martinez et al., 2014). Importantly, the stages in RQ naturally have unequal significance—early codewords capture most of the energy while later stages refine the residual—which enables adaptive computation by truncating decoding to a prefix of stages under a compute or latency budget.

Recently, VQ has started to be explored for KV cache compression in LLM inference. Coupled Quantization (CQ) can be interpreted as a PQ-style scheme: it partitions KV channels into small contiguous blocks and jointly quantizes each block with a per-block codebook (Zhang et al., 2024). VQLLM instead adopts RQ to reduce reconstruction error by sequentially encoding the residual (Kumar, 2024). Meanwhile, AnTKV (Li et al., 2025) focuses on mitigating the impact of outliers by using weighted $k$-means and preserving anchor (important) tokens without compression. Despite these differences, most VQ pipelines in prior work rely on $K$-means as a core primitive for learning codebooks under an $\ell_2$ reconstruction objective.

In this paper, we identify a subtle failure mode of standard $K$-means under an $\ell_2$ reconstruction objective: in high dimensions, centroid averaging can compromise direction and magnitude in a way that is easy to miss yet consequential for reconstruction quality. This effect is especially likely when high-dimensional vectors are handled in multi-stage schemes such as RQ. To mitigate this issue, we propose Gain–Shape $K$-means (GSKM), a drop-in replacement for standard $K$-means. Building on GSKM, we develop *Gain-Shape Residual Quantization* (GSRQ), a RQ pipeline for KV cache compression and show that it improves reconstruction quality and downstream accuracy compared to recent VQ/RQ baselines.

In summary, our contributions are as follows:

- We characterize an often-overlooked degradation mechanism of standard $K$-means that arises when clustering high-dimensional, weakly structured vectors, and explain why multi-stage quantizers (e.g., RQ/AQ) can exacerbate it.

- We introduce GSKM, a drop-in replacement for standard $K$-means that better preserves direction while sometimes even reducing gain error and $\ell_2$ distortion. Unlike many prior approaches that exploit structured patterns, GSKM is most effective in the opposite regime—when representations are weakly structured or random-like—making it complementary to structure-exploiting methods.

- We build GSRQ by integrating GSKM with a robust gradient-based weighting scheme within a RQ pipeline for KV cache quantization. This combined design yields substantial improvements over state-of-the-art VQ/RQ baselines on LLM inference workloads.

## 2. Related Works

**Gain-Shape Decomposition.** Gain–shape vector quantization (GSVQ), also referred to as shape–gain quantization, decomposes a vector into its magnitude (gain) and direction (shape), which are encoded using separate gain and shape codebooks. This formulation can be viewed as a special case of product vector quantization, where the overall codebook is formed by the Cartesian product of lower-dimensional component codebooks (Sabin & Gray, 1984; Gray, 1984; Gersho & Gray, 1992; Canta et al., 1996). This separation is motivated by both computational and statistical considerations. Computationally, product codes reduce storage and search complexity: while the effective codebook size grows multiplicatively, memory usage and encoding complexity scale only additively with the component codebook sizes (Sabin & Gray, 1984; Gray, 1984; Gray & Neuhoff, 2002). Statistically, gain and shape often carry different types of information. In waveform and signal coding, the gain spans a wide dynamic range but carries relatively little information, whereas the shape lies on a unit sphere and captures the essential structural content of the signal, justifying independent quantization and separate bit allocation for the two components (Gray, 1984; Canta et al., 1996). In high-dimensional text or embedding data, the norm can be less informative than angular similarity, motivating spherical $k$-means and related methods that discard gain and cluster on the unit sphere (Dhillon & Modha, 2001; Banerjee et al., 2005). However, discarding gain can be undesirable for KV cache compression, where magnitude still affects $\ell_2$ reconstruction and downstream attention behavior. In contrast, our use of gain–shape is not a product-code construction with separate gain and shape codebooks. Instead, we use gain–shape to explicitly separate direction and magnitude within the $K$-means updates, retaining both factors while reducing the shrinkage–alignment coupling under the $\ell_2$ objective in high dimensions. Empirically, this yields more reliable directional matching while retaining magnitude information, unlike spherical clustering.

## 3. Preliminaries

**The Curse of Dimensionality.** A conventional vector quantizer maps a $D$-dimensional vector $x \in \mathbb{R}^D$ to the nearest codeword in a codebook $\mathcal{C}$. To maintain a target

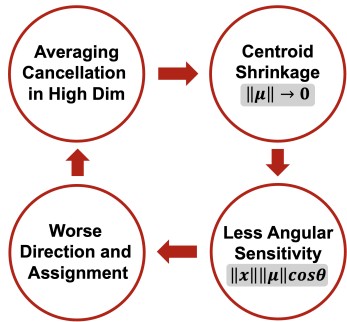

*Figure 1.* **Illustration of a hypothesized vicious cycle in high-dimensional $\ell_2$-based K-Means.** Averaging cancellation induces centroid shrinkage, which reduces angular sensitivity and leads to poorer direction and assignment, further reinforcing cancellation.

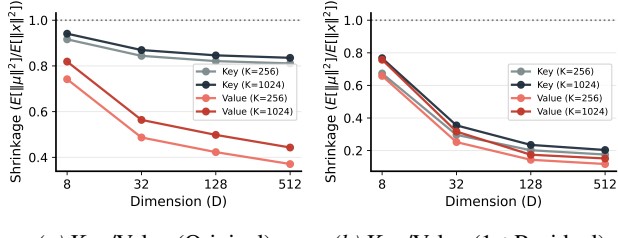

*(a)* Key/Value (Original)  *(b)* Key/Value (1st Residual)

*Figure 2.* **Centroid shrinkage in subspace K-means on LLaMA-3-8B KV cache (Lower means stronger shrinkage).** As the subspace dimension $D$ increases, shrinkage becomes progressively more severe; it is consistently stronger for *values* than *keys*, and is most pronounced for *residual* values (right).

bit-rate of $b$ bits per dimension, the codebook size must scale as $K = 2^{bD}$, leading to a parameter complexity of $\mathcal{P}_{VQ} \approx D \cdot 2^{bD}$. For high-dimensional vectors, this exponential growth renders standard VQ computationally intractable and memory-prohibitive.

**Product Quantization.** Product Quantization (PQ) addresses this intractability by decomposing the high-dimensional space into $M$ orthogonal subspaces and quantizing each sub-vector independently. This decomposition reduces the storage complexity to $\mathcal{P}_{PQ} \approx D \cdot 2^{bD/M}$. While computationally efficient, PQ inherently assumes independence between subspaces, thereby ignoring correlations and limiting reconstruction fidelity.

**Residual Quantization.** Residual Quantization (RQ) refines the approximation by representing the vector $x$ as a sum of $L$ codewords, $x \approx \sum_{l=1}^{L} \mathcal{C}_l[i_l]$, where each stage sequentially quantizes the residual error of the previous stage. This iterative process expands the representational capacity without increasing dimensionality, reducing the complexity to $\mathcal{P}_{RQ} \approx L \cdot D \cdot 2^{bD/L}$. Consequently, RQ effectively mitigates the curse of dimensionality by reducing the complexity exponent by a factor of $L$.

## 4. Motivation: When $\ell_2$-Optimal Centroids Lose Scale and Direction

VQ and K-means typically represent a data vector $x \in \mathbb{R}^D$ using a centroid $\mu \in \mathbb{R}^D$ and minimize squared euclidean distortion $\|x - \mu\|_2^2$. Although convenient, this objective implicitly couples two geometrically distinct factors: (i) the *magnitude* (norm) of the vector and (ii) its *direction* (angle). By the law of cosines,

$$\|x - \mu\|_2^2 = \|x\|_2^2 + \|\mu\|_2^2 - 2\|x\|_2 \|\mu\|_2 \cos\theta, \quad (1)$$

where $\theta$ is the angle between $x$ and $\mu$. Equation (1) makes the coupling explicit: the angular term scales with

$\|x\|_2\|\mu\|_2$, so when $\|\mu\|_2$ is small, improvements in $\cos\theta$ have a diminished effect on $\ell_2$ distortion. This structure also allows the objective to be reduced by shrinking $\|\mu\|_2$ even when directional alignment does not improve.

Centroid norms are smaller than the average norm of the assigned vectors unless samples are perfectly aligned, and in high dimensions they can become substantially smaller due to mean cancellation under angular dispersion. Formally, for a cluster $C$, the $\ell_2$-optimal centroid is the arithmetic mean of the assigned points, $\mu^\star = \mathbb{E}[x \mid x \in C]$, which satisfies

$$\|\mu^\star\|_2 = \|\mathbb{E}[x \mid x \in C]\|_2 \leq \mathbb{E}[\|x\|_2 \mid x \in C], \quad (2)$$

with equality only when samples are perfectly aligned. As dimensionality increases, within-cluster directions tend to be more dispersed and nearly orthogonal (Hall et al., 2005), making cancellation in the mean more likely and the resulting shrinkage more pronounced.

While dispersion alone induces compromise directions and shrinkage, we hypothesize that $\ell_2$-driven centroid updates can further reinforce directional degradation in the high-dimensional, under-capacity regime. In particular, as $\|\mu^\star\|_2 \to 0$, the objective becomes less sensitive to improving $\cos\theta$ because the angular term in (1) scales with $\|\mu\|_2$. As a result, shrinkage weakens the pressure to maintain or correct directional alignment, allowing directional compromise to persist and potentially amplify.

This hypothesized mechanism is particularly relevant in RQ pipelines. These methods typically encode *full-dimensional* vectors at each stage (large $D$), and distribute a fixed overall budget across multiple stages. As a result, each stage often operates with a relatively small codebook size, i.e., in an *under-capacity* regime where a single centroid must cover a broad, high-dimensional residual distribution and may implicitly average multiple nearly-orthogonal modes. This setting amplifies centroid shrinkage and directional compromise, motivating a clustering primitive that explicitly

preserves directional structure under high-dimensional angular dispersion. Figure 1 illustrates the detrimental cycle.

Figure 2 shows an example of this regime in KV cache quantization. We split per-layer key/value vectors into $D$-dimensional subspaces and run K-means independently per subspace (shown for $K \in \{256, 1024\}$), then report the layer-averaged shrinkage ratio $\mathbb{E}\|\mu\|_2^2 / \mathbb{E}\|x\|_2^2$ (lower means stronger shrinkage). As $D$ grows, each centroid must represent increasingly high-dimensional variation with a fixed (and thus effectively smaller) per-dimension capacity, making mean cancellation more likely and exacerbating shrinkage. Values exhibit stronger shrinkage than keys, consistent with findings in (Liu et al., 2024) that keys tend to be more structured than values. Residuals exhibit even stronger shrinkage, suggesting that their structure is largely diminished relative to the original vectors.

## 5. Methodology

### 5.1. Overview

In *Gain–Shape Residual Quantization (GSRQ)*, we combine Product Quantization (PQ) and Residual Quantization (RQ) to balance codebook tractability with reconstruction fidelity. PQ partitions each vector into $M$ disjoint subspaces, reducing the effective dimensionality per codebook, while RQ progressively refines the approximation by sequentially quantizing residuals over $L$ stages within each subspace. The hyperparameters $(M, L)$ provide a practical knob for controlling codebook granularity and can be chosen to improve cache locality and reduce lookup overhead at decoding time.

Most VQ methods, including PQ and RQ, rely on $K$-means (KM) as a core primitive for learning codebooks. In GSRQ, we replace every occurrence of KM used for codebook learning with Gain–Shape $K$-means (GSKM), introduced in Subsection 5.2. A common extension of KM is weighted $K$-means, which incorporates per-sample weights to emphasize more important or reliable points when learning codebooks. We also extend GSKM in the same manner and use a weighted GSKM variant, with the weighting scheme introduced in Section 5.3.

### 5.2. Gain–Shape K-means (GSKM)

GSKM is motivated by a practical failure mode of standard $\ell_2$ $K$-means in high dimensions: when cluster means shrink toward the origin due to averaging cancellation, the inferred direction can become numerically fragile and less angle-sensitive, which in turn degrades downstream use of the prototypes. To mitigate this, we decouple direction and magnitude by maintaining an explicit unit-norm *shape* variable and updating it with a magnitude-invariant rule. This design provides a robust, tuning-free default for estimating

cluster directions, while still using a simple closed-form update for the gain.

**Gain–shape parameterization.** For each cluster $k \in \{1, \ldots, K\}$, we represent the centroid as

$$\mu_k = g_k s_k, \qquad g_k \geq 0, \ \ \|s_k\|_2 = 1, \qquad (3)$$

where $g_k$ is a scalar *gain* and $s_k$ is a unit-norm *shape* (direction). We introduce gain–shape as a *re-parameterization* to stabilize optimization against the $\ell_2$ shrinkage–alignment coupling in high dimensions, not as a coding heuristic. Consequently, although $g_k$ and $s_k$ are optimized separately, they jointly define a single centroid $c_k = g_k s_k$ and can be viewed as an implicit single codebook.

**Assignment.** The squared distance decomposes as

$$\|x - g_k s_k\|_2^2 = \|x\|_2^2 + g_k^2 - 2g_k(x^\top s_k), \qquad (4)$$

making explicit that angular alignment enters through the projection $x^\top s_k$, while magnitude contributes a quadratic penalty $g_k^2$. Given current $(g_k, s_k)$, each $x_i$ is assigned by

$$k_i^\star = \arg\min_k \|x_i - g_k s_k\|_2^2 = \arg\max_k \left(2g_k x_i^\top s_k - g_k^2\right), \qquad (5)$$

dropping the constant $\|x_i\|_2^2$. Let $\mathcal{I}_k = \{i \mid k_i^\star = k\}$ denote the assigned set.

**Shape update (magnitude-invariant direction).** We first update the direction using per-sample normalized vectors

$$y_i \leftarrow \frac{x_i}{\|x_i\|_2}, \qquad (6)$$

and set the shape by normalizing the within-cluster mean:

$$s_k \leftarrow \frac{\frac{1}{|\mathcal{I}_k|} \sum_{i \in \mathcal{I}_k} y_i}{\left\| \frac{1}{|\mathcal{I}_k|} \sum_{i \in \mathcal{I}_k} y_i \right\|_2}. \qquad (7)$$

This update is the closed-form maximizer of an angular alignment objective $\max_{\|s\|_2=1} \sum_{i \in \mathcal{I}_k} y_i^\top s$, yielding a parameter-free, magnitude-invariant estimate of the cluster direction.

**Gain update (mean projection).** Given the updated direction $s_k$, minimizing (4) over $g_k$ yields the mean projection onto $s_k$,

$$g_k \leftarrow \max\left(0, \ \frac{1}{|\mathcal{I}_k|} \sum_{i \in \mathcal{I}_k} x_i^\top s_k\right), \qquad (8)$$

where the nonnegativity clamp enforces the gain–shape identification ($\mu_k = g_k s_k$ with $g_k \geq 0$) and avoids degenerate sign flips in practice.

**Algorithm 1** Gain–Shape K-means (GSKM)

**Require:** Data $\mathcal{D} = \{x_i\}_{i=1}^N$, number of clusters $K$, max iterations $T$
**Ensure:** Gains $\{g_k\}_{k=1}^K$, shapes $\{s_k\}_{k=1}^K$ (centroids $\mu_k = g_k s_k$)
 1: **Initialize:** choose $\{s_k\}_{k=1}^K$ on the unit sphere.
 2: Initialize gains by (8).
 3: **for** $t = 1$ **to** $T$ **do**
 4:     // Assignment
 5:     **for** $i = 1$ **to** $N$ **do**
 6:        $k_i \leftarrow \arg\max_{k \in \{1,\ldots,K\}} \left( 2g_k x_i^\top s_k - g_k^2 \right)$
 7:     **end for**
 8:     // Update
 9:     **for** $k = 1$ **to** $K$ **do**
10:        $\mathcal{I}_k \leftarrow \{i \mid k_i = k\}$
11:        $y_i \leftarrow x_i / \|x_i\|_2$ for all $i \in \mathcal{I}_k$
12:        $s_k \leftarrow \left( \frac{1}{|\mathcal{I}_k|} \sum_{i \in \mathcal{I}_k} y_i \right) \Big/ \left\| \frac{1}{|\mathcal{I}_k|} \sum_{i \in \mathcal{I}_k} y_i \right\|_2$
13:        $g_k \leftarrow \max\left( 0, \frac{1}{|\mathcal{I}_k|} \sum_{i \in \mathcal{I}_k} x_i^\top s_k \right)$
14:     **end for**
15:     **if** assignments $\{k_i\}$ unchanged **then**
16:        **break**
17:     **end if**
18: **end for**
19: **return** $\{g_k\}_{k=1}^K$, $\{s_k\}_{k=1}^K$

**Practical note on objective and convergence.** Because the shape update (7) optimizes an angular criterion while the assignment and gain updates follow the raw $\ell_2$ distortion structure, Algorithm 1 is not an exact block coordinate descent procedure for a single $\ell_2$ objective, and we do not claim a monotonic decrease guarantee. We instead treat it as a practical heuristic: empirically, the magnitude-invariant direction update substantially improves robustness of direction estimation in high-dimensional regimes and translates to consistent improvements on downstream application metrics. Importantly, the method introduces no additional hyperparameters, providing a strong default that avoids tuning overhead while delivering reliable gains in practice. The full procedure is summarized in Algorithm 1.

**Implementation details.** In practice, we handle corner cases with standard safeguards: if $\mathcal{I}_k = \emptyset$ we keep $(g_k, s_k)$ unchanged (or reinitialize the cluster), and we compute $y_i = x_i/(\|x_i\|_2 + \epsilon)$ with a small $\epsilon$ to avoid division by zero; if the mean direction $\frac{1}{|\mathcal{I}_k|} \sum_{i \in \mathcal{I}_k} y_i$ is numerically close to zero, we also keep the previous $s_k$.

**Complexity.** GSKM has the same asymptotic complexity as K-means: assignment costs $O(NKD)$ dot products per iteration, and the updates cost $O(ND)$ accumulations.

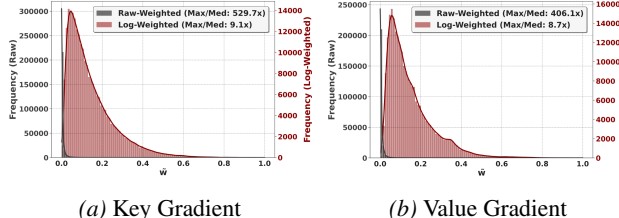

*(a) Key Gradient*      *(b) Value Gradient*

*Figure 3.* **Distribution of Key and Value Gradients.** We compare the distributions of raw L2 norms ($\tilde{w} = \|\mathbf{g}\|_2$, grey) and log-processed norms ($\tilde{w} = \log(1 + \lambda\|\mathbf{g}\|_2)$, red) for (a) Key and (b) Value Gradients, where the maximum values are scaled to 1. Raw gradient norms are heavy-tailed, whereas log-smoothing compresses their dynamic range.

### 5.3. Gradient-Weighted GSKM

Standard Euclidean clustering in KV cache compression minimizes average distortion but ignores the varying sensitivity of the global loss $\mathcal{L}$ to quantization errors across tokens. Following the sensitivity-based weighting of SqueezeLLM (Kim et al., 2023), we adapt this approach to our setup.

However, directly utilizing raw gradient magnitudes can be suboptimal, as the weighting distribution is often dominated by *highly salient tokens* that exhibit extremely large gradients (Xiao et al., 2024a). To make the weighting robust to such outliers, we apply a logarithmic transform:

$$w_i = \log\left(1 + \lambda\|\nabla_{x_i}\mathcal{L}\|_2\right), \tag{9}$$

where $\lambda$ normalizes gradient magnitudes relative to the sample median. We apply this weighting consistently at every residual stage, enforcing the same sensitivity prior throughout the hierarchy.

Figure 3 shows that raw gradient norms are heavy-tailed (max/median $> 400\times$ for both keys and values), whereas log-smoothing reduces this ratio to below $10\times$. This stabilization yields lower perplexity than using raw gradient-norm weighting. We provide an ablation for selecting $\lambda$ in Appendix B.

## 6. Experiments

### 6.1. GSKM Evaluation

**Baselines and metrics.** Across all evaluations, we compare standard $K$-means (KM) against our GSKM. Given a dataset of vectors $\{x_i\}_{i=1}^N \subset \mathbb{R}^D$, each method learns a codebook of size $K$ and reconstructs each vector by its assigned prototype, yielding $\hat{x}_i$. We report: (i) mean-squared error MSE $= \mathbb{E}[\|x - \hat{x}\|_2^2]$, (ii) average gain error $\mathbb{E}\left[\left|\|x\|_2 - \|\hat{x}\|_2\right|\right]$, and (iii) cosine similarity $\mathbb{E}[\cos(x, \hat{x})]$.

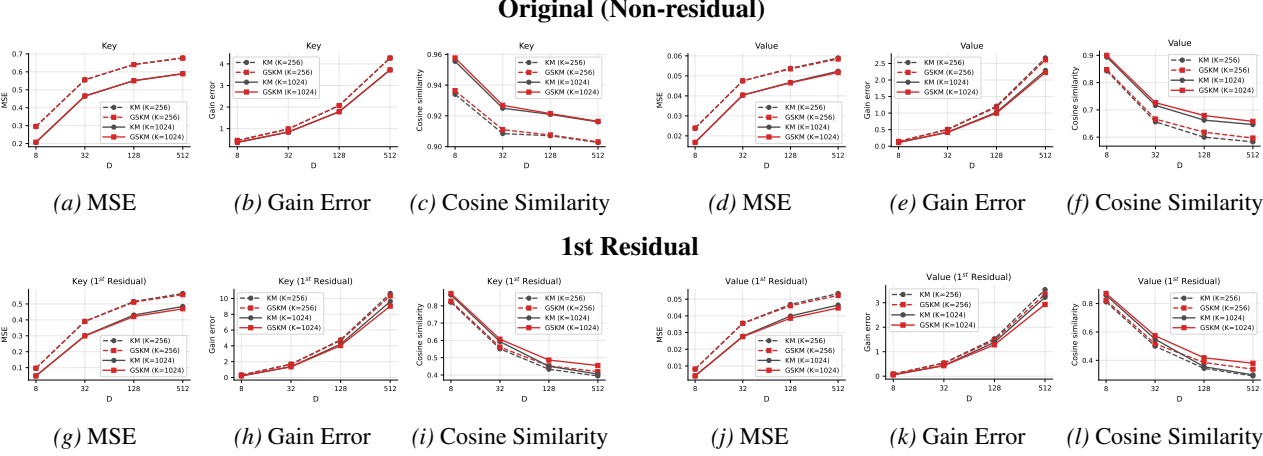

**Figure 4. Random normal sweeps.** *(a-c)*: dimension sweep (vary $D$ with fixed $K$=2048). *(d-f)*: capacity sweep (vary $K$ with fixed $D$=256). Across both sweeps, GSKM improves directional alignment (higher cosine similarity) and reduces gain error relative to KM, often translating into lower MSE. The gap is largest in the under-capacity regime.

**Figure 5. KV cache reconstruction (LLaMA-3-8B, Wikitext-2).** KM vs. GSKM on keys/values for original activations (top) and first residuals (bottom) at $K \in \{256, 1024\}$. GSKM improves cosine similarity and gain error (often, even MSE), with more pronounced on values; in residuals, $K$=256 GSKM can surpass $K$=1024 KM in directional alignment.

**Evaluation on Random Gaussian.** We first evaluate KM and GSKM on a controlled synthetic distribution with i.i.d. samples $x \sim \mathcal{N}(0, I_D)$. For each setting, we quantize $N$=10,000 samples with codebook size $K$ and compute the metrics above. Figure 4(*a-c*), GSKM consistently improves the directional fidelity of the reconstruction, yielding higher cosine similarity than KM. Notably, this directional advantage does not come at the expense of magnitude: GSKM also reduces gain error, which in turn translates into lower overall MSE. This trend becomes more apparent as $D$ increases, where vanilla KM is more prone to directional compromise under high-dimensional averaging, while GSKM preserves a better gain–shape decomposition. Figure 4(*d-f*) further highlights the regime where the difference is most pronounced. When $K$ is small (the under-capacity regime), KM exhibits a sharp degradation in direction and gain, whereas GSKM remains substantially more stable, leading to a significantly larger gap in cosine similarity. As $K$ increases and capacity becomes sufficient, the two methods converge, consistent with the expectation that the benefit of GSKM is strongest when representational capacity is limited.

**Evaluation on KV cache.** We next evaluate on real KV cache activations from LLaMA-3-8B using Wikitext-2. For each of keys and values, we collect $N$=243,035 vectors across layers. We concatenate attention heads to form 1024-dimensional vectors, then partition each vector into $D$-dimensional subspaces and run KM or GSKM independently per subspace. We obtain the first residual by subtracting the KM reconstruction from the original activations. We report metrics averaged across layers and subspaces. Figure 5 shows a similar pattern on original KV activations as in the random-data sweep: GSKM improves directional alignment and gain preservation, often even reducing MSE. The effects are more pronounced for values, which are less structured than keys. In the first-residual setting, GSKM exhibits a more pronounced improvement than KM: at $K$=256, GSKM achieves better directional alignment than KM with $K$=1024 for both keys and values.

### 6.2. KV Compression Evaluation

**Models and Datasets.** We evaluate the efficiency of our proposed method, GSRQ on KV cache activations, across a diverse set of large language models (LLMs) and benchmarks. For standard perplexity evaluations, we utilize the base versions of LLaMA-2-7B (Touvron et al., 2023), LLaMA-3-8B (Dubey et al., 2024), and Mistral-

*Table 1.* **Perplexity comparison.** Lower is better. *FP16* denotes the full-precision baseline. GSRQ consistently achieves the lowest perplexity across three LLMs and both WikiText-2 and C4 at matched bit budgets.

| METHOD | BIT | LLAMA-2-7B | | LLAMA-3-8B | | MISTRAL-7B | |
| | | WIKITEXT-2 | C4 | WIKITEXT-2 | C4 | WIKITEXT-2 | C4 |
|---|---|---|---|---|---|---|---|
| *FP16* | **16** | 5.12 | 6.63 | 5.54 | 7.10 | 4.73 | 5.66 |
| CQ | **2** | 5.42 | 7.23 | 6.09 | 18.71 | 5.11 | 6.17 |
| ANTKV | **2** | 5.51 | 7.45 | 6.10 | 16.96 | 5.08 | 6.18 |
| **GSRQ** | **2** | **5.34** | **6.88** | **5.91** | **7.43** | **4.98** | **5.80** |
| CQ | **1** | 7.75 | 12.49 | 9.56 | 81.74 | 7.25 | 9.89 |
| ANTKV | **1** | 7.92 | 13.01 | 9.62 | 74.47 | 7.32 | 10.51 |
| **GSRQ** | **1** | **6.11** | **7.93** | **7.20** | **8.92** | **6.11** | **6.53** |
| CQ | **0.75** | 8.39 | 14.32 | 11.18 | 72.05 | 7.64 | 11.72 |
| ANTKV | **0.75** | 8.21 | 14.27 | 10.41 | 66.28 | 7.41 | 11.72 |
| **GSRQ** | **0.75** | **6.81** | **8.99** | **8.46** | **10.73** | **7.14** | **7.38** |
| CQ | **0.375** | 14.82 | 33.59 | 22.80 | 103.5 | 13.20 | 26.34 |
| ANTKV | **0.375** | 13.37 | 30.51 | 17.70 | 103.5 | 11.65 | 23.98 |
| **GSRQ** | **0.375** | **9.42** | **14.19** | **12.77** | **20.17** | **10.07** | **11.08** |

*Table 2.* **Few-shot performance comparison on common sense reasoning and math benchmarks.** Higher is better. GSRQ consistently outperforms VQLLM at matched bit budgets, and even at 0.75 bit it achieves higher average accuracy than VQLLM at 1 bit, highlighting strong sub-1-bit performance.

| METHOD | BIT | ARC-C | MMLU | TRUTHFULQA | WINOGRANDE | HELLASWAG | PIQA | MATHQA | AVERAGE |
|---|---|---|---|---|---|---|---|---|---|
| FP16 | 16 | 62.37 | 66.67 | 52.49 | 76.48 | 78.84 | 78.94 | 43.45 | 65.60 |
| VQLLM | 2 | 57.76 | 60.68 | 51.05 | 69.30 | 75.23 | 76.88 | 39.16 | 61.43 |
| **GSRQ** | 2 | 59.30 | 63.46 | 50.12 | 73.16 | 76.38 | 78.56 | 41.07 | **63.15** |
| VQLLM | 1 | 37.03 | 28.29 | 44.25 | 53.12 | 51.65 | 72.69 | 25.23 | 44.60 |
| **GSRQ** | 1 | 52.05 | 54.32 | 48.67 | 68.98 | 69.56 | 76.06 | 34.84 | **57.78** |
| **GSRQ** | **0.75** | 45.22 | 46.50 | 47.26 | 63.54 | 63.74 | 75.68 | 31.76 | **53.38** |

7B-v0.1 (Jiang et al., 2023), measuring performance on the WikiText-2 (Merity et al., 2017) and C4 (Raffel et al., 2020) datasets. To further assess the model's capabilities in long-context understanding and mathematical reasoning, we evaluate LLaMA-3-8B-Instruct on LongBench (Bai et al., 2023) and, alongside Mistral-7B-Instruct-v0.1, on the GSM8K (Cobbe et al., 2021) benchmark. For calibration in these instruction-tuned tasks, we utilize the SlimPajama dataset (Soboleva et al., 2023).

**Baselines.** We benchmark our approach against state-of-the-art KV cache quantization methods, including CQ (Zhang et al., 2024), AnTKV (Li et al., 2025), KIVI (Liu et al., 2024), and VQLLM (Kumar, 2024). AnTKV-1% (Li et al., 2025) keeps a subset of important tokens in full precision, and we do not include it as a baseline because anchor-token selection is an orthogonal technique that can be used in conjunction with other KV-compression methods.

**Configurations.** We evaluate 2, 1, 0.75, and 0.375 bits per activation (BPA). With a fixed codebook size $K=256$, we

set the subspace dimension $D$ and the number of residual stages $R$ as follows. For 2/1/0.75 BPA, we use $D=128$ with $R=32/16/12$, respectively. For 0.375 BPA, we use $D=256$ and $R=12$. The average bitwidth is calculated as $\frac{R \log_2(K)}{D}$. Codebooks are calibrated using 128 training samples with a sequence length of 2048, while evaluation is performed with a maximum sequence length of 8K on a single NVIDIA A100 (80GB) GPU.

**Perplexity Evaluation.** Table 1 reports perplexity (PPL; lower is better) on the Wikitext-2 and C4 validation sets for three LLMs (LLaMA-2-7B, LLaMA-3-8B, and Mistral-7B). Across all models and bit-rates where baselines are available, GSRQ consistently achieves lower PPL than competing methods. The improvement becomes increasingly pronounced as the bit-rate decreases: in the low-bit regime (0.75 and 0.375 BPA), baselines often enter an under-capacity setting where each stage must represent broad variation with a small codebook, leading to rapidly deteriorating PPL. In contrast, GSRQ degrades more gracefully, aligning

*Table 3.* **Performance comparison on LongBench tasks.** GSRQ consistently attains higher accuracy compared to VQLLM and KIVI.

| Method | Bit | NarrativeQA | Qasper | MultifieldQA | QMSum | MuSiQue | TriviaQA | SAMsum | Average |
|---|---|---|---|---|---|---|---|---|---|
| FP16 | 16 | 23.94 | 39.66 | 41.52 | 23.03 | 22.24 | 90.11 | 41.64 | 40.30 |
| KIVI-2 | 2.25 | 23.00 | 34.97 | 38.96 | 22.37 | 20.68 | 90.05 | 41.12 | 38.73 |
| VQLLM | 2.00 | 17.41 | 32.65 | 38.55 | 22.07 | 19.22 | 87.42 | 39.52 | 36.69 |
| **GSRQ** | **2.00** | 21.20 | 36.92 | 40.79 | 22.81 | 24.33 | 89.20 | 42.12 | **39.62** |
| KIVI-1 | 1.25 | 2.18 | 3.12 | 5.06 | 7.29 | 2.43 | 15.91 | 5.82 | 5.97 |
| VQLLM | 1.00 | 1.26 | 4.04 | 8.76 | 16.17 | 2.12 | 35.40 | 11.69 | 11.34 |
| **GSRQ** | **1.00** | 18.59 | 28.73 | 30.42 | 20.83 | 17.08 | 83.41 | 35.72 | **33.54** |
| **GSRQ** | **0.75** | 19.71 | 18.31 | 28.92 | 20.12 | 13.51 | 76.3 | 31.19 | **29.72** |

*Table 4.* **Performance comparison on GSM8K benchmark.** The exact match accuracy (%) for LLaMA-3-8B-Instruct and Mistral-7B-Instruct-v0.1.

| | | GSM8K | |
|---|---|---|---|
| Method | Avg. bit | LLaMA-3-8B | Mistral-7B |
| FP16 | 16 | 76.19 | 33.43 |
| VQLLM | 2.00 | 55.65 | 27.90 |
| **GSRQ** | **2.00** | **65.73** | **28.13** |
| VQLLM | 1.00 | 2.27 | 3.94 |
| **GSRQ** | **1.00** | **26.16** | **13.65** |

*Table 5.* **Performance comparison on MATH-500 and RULER benchmarks for LLaMA-3-8B-Instruct.** GSRQ preserves stronger mathematical reasoning and long-context performance than existing KV cache quantization baselines at matched bit budgets.

| Method | BPA | MATH-500 | RULER |
|---|---|---|---|
| FP16 | 16 | 20.4 | 96.0 |
| KIVI | 2 | 15.8 | 94.0 |
| VQLLM | 2 | 13.4 | 94.0 |
| **GSRQ** | **2** | **19.0** | **96.0** |
| KIVI | 1 | 0.8 | 13.0 |
| VQLLM | 1 | 0.2 | 20.0 |
| **GSRQ** | **1** | **14.2** | **72.0** |

*Table 6.* **Ablation study of GSRQ (Log-Weighted-GSKM).** On LLaMA-3-8B (0.375-bit) with WikiText-2, GSKM provides the largest perplexity gain within GSRQ.

| | Perplexity | | |
|---|---|---|---|
| Method | Both | Key | Value |
| Standard KM | 19.50 | 7.28 | 12.30 |
| Unweighted-GSKM | 12.77 | 6.87 | 8.88 |
| Raw-Weighted-GSKM | 13.57 | 6.90 | 9.35 |
| **Log-Weighted-GSKM** | **12.49** | **6.82** | **8.78** |

tasks. We conduct the evaluation on ARC-C (25-shot) (Clark et al., 2018), MMLU (5-shot) (Hendrycks et al., 2020), TruthfulQA (0-shot) (Lin et al., 2022), Winogrande (5-shot) (Sakaguchi et al., 2021), HellaSwag (10-shot) (Zellers et al., 2019), PIQA (0-shot) (Bisk et al., 2020), MathQA (5-shot) (Amini et al., 2019), and GSM8K (5-shot) (Cobbe et al., 2021). Table 2 first compares GSRQ against VQLLM, a recent RQ baseline that augments vanilla RQ with Exponential Moving Average (EMA) codebook learning and non-contiguous grouping. Despite using a simpler modification—a drop-in substitution of $K$-means with a weighted GSKM within the vanilla RQ pipeline—GSRQ substantially improves accuracy over VQLLM. Notably, GSRQ at 0.75-bit already surpasses VQLLM at 1-bit. Table 3 reports results on LongBench. GSRQ at 0.75-bit outperforms both VQLLM at 1-bit and KIVI-1, indicating strong robustness as the quantization budget decreases. We further verify architectural generalization by extending our evaluation to Mistral-7B-Instruct on the GSM8K benchmark, as reported in Table 4. The results demonstrate consistent robustness across models, notably mitigating the severe performance collapse observed in baselines in the 1-bit regime. Table 5 further shows that GSRQ preserves stronger mathematical reasoning and long-context performance on MATH-500 and RULER.

with our earlier finding that GSKM is particularly effective under under-capacity and weakly structured representations. Finally, the advantage of GSRQ is more evident on C4 than on Wikitext-2. While both datasets show consistent gains, C4 exhibits substantially larger gaps between GSRQ and prior methods, especially at 1-bit and below, suggesting that robustness to the low-bit under-capacity regime is critical for maintaining performance on more diverse corpora.

**Benchmarks Evaluation.** We evaluate downstream quality using the LM Evaluation Harness (Gao et al., 2024) on commonsense and math benchmarks, as well as LongBench

### 6.3. GSRQ Ablation

We analyze the contribution of each component using LLaMA-3-8B evaluated on the WikiText-2 dataset, as de-

*Table 7.* **Few-shot performance comparison on commonsense and math benchmarks for Qwen3-8B.** GSRQ improves average accuracy over VQLLM at matched low-bit budgets.

| METHOD | BIT | ARC-C | MMLU | TRUTHFULQA | WINOGRANDE | HELLASWAG | PIQA | MATHQA | AVERAGE |
|---|---|---|---|---|---|---|---|---|---|
| FP16 | 16 | 26.79 | 26.37 | 45.79 | 50.43 | 31.94 | 62.02 | 24.46 | 38.26 |
| VQLLM | 2 | 22.18 | 25.88 | 51.64 | 50.67 | 25.47 | 56.86 | 22.35 | 36.44 |
| **GSRQ** | **2** | 23.89 | 29.91 | 45.65 | 53.91 | 29.47 | 73.23 | 29.18 | **40.75** |
| VQLLM | 1 | 22.70 | 24.11 | 51.09 | 50.04 | 25.72 | 53.92 | 20.30 | 35.41 |
| **GSRQ** | **1** | 21.50 | 26.19 | 48.82 | 54.14 | 27.47 | 66.38 | 24.52 | **38.43** |
| **GSRQ** | **0.75** | 22.95 | 25.87 | 50.51 | 51.85 | 26.86 | 65.40 | 22.48 | **37.99** |

*Table 8.* **Performance comparison on LongBench tasks for Qwen3-8B.** GSRQ achieves stronger average long-context performance than VQLLM under aggressive KV cache compression.

| METHOD | BIT | NARRATIVEQA | QASPER | MULTIFIELDQA | QMSUM | MUSIQUE | TRIVIAQA | SAMSUM | AVERAGE |
|---|---|---|---|---|---|---|---|---|---|
| FP16 | 16 | 22.90 | 47.39 | 54.22 | 22.60 | 30.31 | 88.04 | 39.67 | 43.59 |
| VQLLM | 2.00 | 23.38 | 42.20 | 46.46 | 22.14 | 21.14 | 77.12 | 36.85 | 38.47 |
| **GSRQ** | **2.00** | 21.96 | 44.88 | 48.24 | 22.67 | 24.42 | 75.30 | 38.27 | **39.39** |
| VQLLM | 1.00 | 6.99 | 15.73 | 24.41 | 18.35 | 3.73 | 11.43 | 4.96 | 12.23 |
| **GSRQ** | **1.00** | 18.56 | 30.28 | 40.77 | 21.48 | 11.88 | 8.17 | 10.35 | **20.21** |
| **GSRQ** | **0.75** | 12.39 | 17.42 | 29.70 | 21.11 | 5.46 | 7.91 | 8.83 | **14.96** |

*Table 9.* **Performance comparison on AIME24 and AIME25 for Qwen3-8B.** GSRQ better preserves competition-level mathematical reasoning than VQLLM at the same 2-BPA setting.

| Method | BIT | AIME24 | AIME25 |
|---|---|---|---|
| FP16 | 16 | 20.0 | 23.3 |
| VQLLM | 2 | 13.3 | 13.3 |
| **GSRQ** | **2** | **20.0** | **16.7** |

*Table 10.* **Perplexity comparison on WikiText-2 and C4 for Qwen3-8B.** Lower is better. GSRQ achieves lower perplexity than VQLLM at matched bit budgets.

| Method | BIT | WikiText-2 | C4 |
|---|---|---|---|
| FP16 | 16 | 8.55 | 8.59 |
| VQLLM | 2 | 9.05 | 9.00 |
| **GSRQ** | 2 | **8.88** | **8.80** |
| VQLLM | 1 | 15.79 | 13.52 |
| **GSRQ** | 1 | **10.22** | **9.84** |

tailed in Table 6. The primary performance leap stems from replacing the standard Euclidean objective with our proposed method, which drastically reduces perplexity from 19.50 to 12.77 (Standard KM → Unweighted-GSKM). While naive gradient weighting (Raw-Weighted) suffers from instability due to outlier gradients, applying our logarithmic smoothing (Log-Weighted) effectively balances the optimization. Consequently, the full method achieves the

best performance, reaching a perplexity of 12.49.

**Robustness Across Models and Tasks.** We further examine whether the benefits of GSRQ extend beyond the main LLaMA and Mistral settings. On Qwen3-8B, Tables 7 and 8 show stronger average performance than VQLLM on few-shot commonsense/math benchmarks and LongBench, while Table 9 shows that GSRQ remains effective on challenging competition-level math problems such as AIME24 and AIME25. Table 10 further confirms this trend in perplexity, where GSRQ achieves lower WikiText-2 and C4 perplexity than VQLLM at matched 2-BPA and 1-BPA settings. Overall, these results support the robustness of GSRQ across model families, task types, and evaluation metrics.

## 7. Conclusion

We introduced Gain–Shape $K$-means (GSKM), a drop-in replacement for standard $\ell_2$ $K$-means for learning vector quantization codebooks. Across random sweeps and real KV cache activations, GSKM improves directional fidelity while maintaining competitive $\ell_2$ distortion. Gain-Shape Residual Quantization (GSRQ) built upon a gradient-weighted extension of GSKM obtains consistent perplexity reductions on Wikitext-2 and C4 across LLaMA-2-7B, LLaMA-3-8B, and Mistral-7B, with larger gains at lower bit-rates. GSRQ also improves downstream accuracy on commonsense/math benchmarks and LongBench compared to recent quantization baselines, demonstrating that GSKM yields substantial practical benefits for KV cache quantization.

## Acknowledgements

This work was supported by the Institute of Information & Communications Technology Planning & Evaluation (IITP) grant funded by the Korea government (MSIT) (No. RS-2025-09942968, AI Semiconductor Innovation Lab, Yonsei University).

## Impact Statement

Our work focuses on improving the efficiency of LLM inference via advanced vector quantization. This research directly contributes to reducing the energy consumption of AI systems and enabling the deployment of large models on resource-constrained edge devices. Ultimately, by lowering the computational energy burden, our method promotes environmentally sustainable AI practices and helps mitigate the growing carbon footprint of large-scale model serving.

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

## A. GSKM vs. Standard K-Means

In this section, we provide a detailed quantitative comparison between the proposed Gain-Shape K-means (GSKM) and the standard Euclidean K-means baseline. As discussed in Section 4, standard K-means is prone to centroid shrinkage in high-dimensional subspaces, particularly when operating in the under-capacity regime (i.e., low bits per activation and high dimensionality per subspace). This phenomenon degrades the angular fidelity of the quantized vectors, leading to higher reconstruction errors.

To empirically validate the efficacy of our proposed method, we compare the perplexity (PPL) of Llama-3-8B on the WikiText-2 validation set across various bit-rates. We sweep the Bits Per Activation (BPA) from 0.375 to 2.0, adjusting the subspace dimension $D$ accordingly.

The quantitative results are detailed in Table 11. We observe that RQ utilizing GSKM consistently achieves lower perplexity than the RQ utilizing standard K-Means across all configurations. Notably, the performance gap is significantly more pronounced for the Value cache compared to the Key cache. This finding empirically confirms that GSKM is particularly effective for representations that are *weakly structured or random-like*—a characteristic typical of Value vectors—where standard Euclidean clustering often fails to capture directional information due to severe centroid shrinkage.

*Table 11.* **Perplexity values.** Comparison between Normal K-Means and GSKM evaluated on the WikiText-2 validation set. Note that this comparison employs unweighted clustering for both methods.

| BPA ($D$) | Both | | Key | | Value | |
|---|---|---|---|---|---|---|
| | Normal KM | **GSKM** | Normal KM | **GSKM** | Normal KM | **GSKM** |
| 0.375 (256) | 19.50 | **12.77** | 7.28 | **6.87** | 12.30 | **8.88** |
| 0.750 (128) | 9.49 | **8.46** | 6.51 | **6.36** | 7.43 | **6.95** |
| 1.000 (128) | 7.95 | **7.20** | 6.18 | **6.05** | 6.80 | **6.42** |
| 2.000 (128) | 6.21 | **5.91** | 5.75 | **5.68** | 5.94 | **5.73** |

## B. Ablation study on Lambda ($\lambda$)

In this section, we investigate the impact of the $\lambda$ hyperparameter on the model's performance. As described in the main paper, we apply a logarithmic transformation to the weights to handle their high dynamic range. To ensure consistent behavior of the transformation, we scale the weights such that their median aligns with a specific target value before applying the $\log(1 + x)$ function. Specifically, the transformation is defined as:

$$\tilde{w} = \log(1 + \lambda w), \quad \text{where} \quad \lambda = \frac{\tau}{\text{median}(w) + \epsilon}, \quad w = \|g\|_2 \tag{10}$$

Here, $g$ denotes the original weights, $w$ is the $\ell_2$-norm of $g$, $\tau$ is the target median, and $\epsilon$ is a small constant for numerical stability.

We evaluate candidate values of $\tau$ in $10^{-2}, 10^{-1}, 10^0, 10^1$, and $10^2$ across different BPA configurations of 0.375 and 0.75. Figure 6 illustrates the results. We observe a clear convex trend where deviation from $\tau = 1.0$ leads to degradation in perplexity. When $\tau$ is too small, the values are compressed into the near-linear region of the log function, reducing its effectiveness. Conversely, when $\tau$ is too large, the saturation effect varies.

## C. Decoding Latency Analysis

Our measurements are obtained from a fully custom PyTorch-based autoregressive decoding implementation utilizing high-performance custom Triton kernels to minimize HBM access overhead. Specifically, we use fused kernels for (i) residual quantization and bit-packing of newly generated KV vectors, and (ii) on-the-fly bit-unpacking, dequantization, RoPE, and attention computation during decoding without materializing intermediate FP16 tensors. All measurements in this section are conducted on a single NVIDIA A100 80GB GPU.

By alleviating memory bandwidth bottlenecks, our method achieves significant layer-wise latency reductions, as shown in Table 12. To directly assess the end-to-end impact, we additionally measured the full autoregressive decoding latency

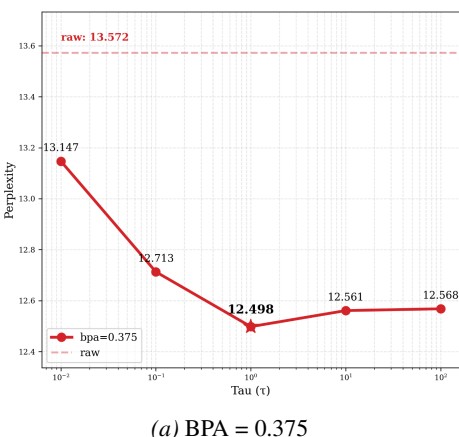

*(a)* BPA = 0.375

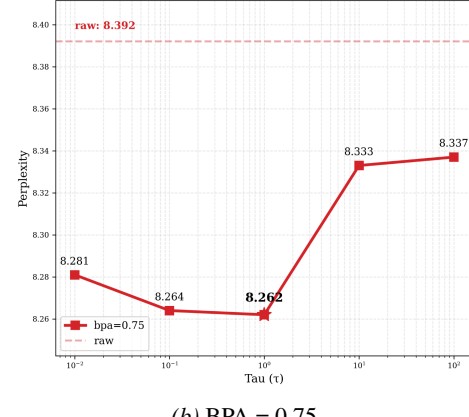

*(b)* BPA = 0.75

*Figure 6.* **Sweeping the target median ($\tau$).** We report the perplexity on the validation set for different BPA configurations. The x-axis represents the target median value used for weight scaling prior to the logarithmic transformation. For comparison, 'raw' denotes the baseline performance where no log smoothing is applied to the weights. The red star ($\star$) denotes the selected configuration ($\tau = 1.0$), which achieves the best performance.

on LLaMA-3-8B under BPA=0.75, using *torch.cuda.Event* timers with synchronization. As reported in Table 13, GSRQ achieves substantial latency reductions at long contexts, delivering a 1.59× to 3.40× speedup in per-token decoding latency over the evaluated range.

Furthermore, as illustrated in Figure 7, our method dramatically reduces the memory footprint. While the FP16 baseline encounters OOM errors at larger batch sizes (Figure 7) and runs out of memory at a 128K context length (Table 13), our method scales efficiently even with limited resources. This substantial reduction in memory usage implies that our approach can effectively support memory-intensive scenarios, such as processing longer context lengths within the same memory budget.

*Table 12.* **Latency comparison.** Layer-wise per-token latency (ms) of FP16 and GSRQ across different batch sizes, measured with a context length of 1024.

| Batch Size | 8 | 16 | 32 |
|---|---|---|---|
| FP16 | 2.3 | 3.9 | 7.9 |
| GSRQ | 1.2 | 1.7 | 2.3 |
| **Speedup** | **1.9×** | **2.3×** | **3.4×** |

*Table 13.* **End-to-end decoding latency per token.** We report the end-to-end per-token decoding latency measured with a batch size of 1 over 100 generated tokens.

| Context length | FP16 (ms) | GSRQ (ms) | Speedup |
|---|---|---|---|
| 8K | 70.68 | 44.48 | **1.59×** |
| 16K | 122.50 | 54.47 | **2.25×** |
| 32K | 253.69 | 83.96 | **3.02×** |
| 64K | 486.73 | 143.29 | **3.40×** |
| 128K | **OOM** | 262.12 | – |

# D. Convergence Analysis

We empirically investigate the convergence behavior of GSKM with respect to the maximum number of iterations to determine the optimal trade-off between computational cost and reconstruction quality. As illustrated in Figure 8, the model demonstrates robust empirical convergence across different hyperparameter settings.

Increasing the maximum iterations up to 40 yields a rapid and consistent reduction in perplexity across all evaluated BPAs. Beyond 40 iterations, the performance improvements become marginal, with the perplexity effectively plateauing in every case. This empirical evidence demonstrates that the algorithm converges efficiently without degradation at higher iterations, suggesting that setting the maximum iteration to around 100 is sufficient to achieve optimal performance while mitigating unnecessary computational overhead.

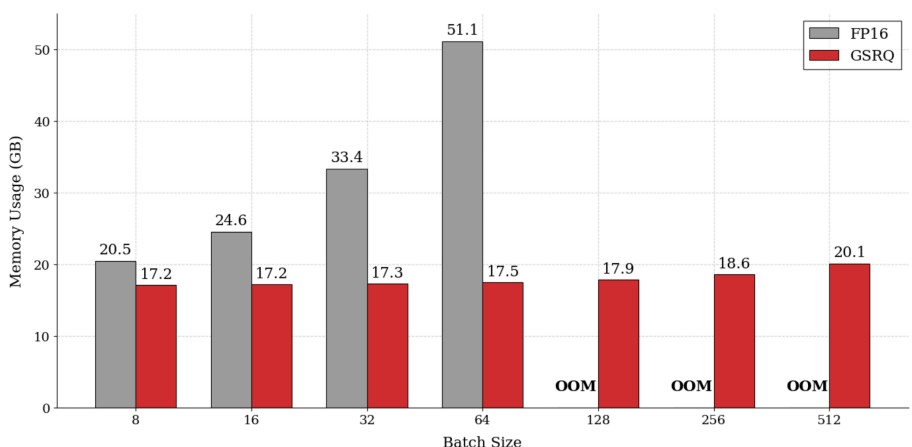

*Figure 7.* **Memory usage during decoding.** Comparison between FP16 and GSKM-0.75bit on LLaMA-3-8B model with a context length of 1K across varying batch sizes. All experiments were performed on a single NVIDIA A100 80GB GPU.

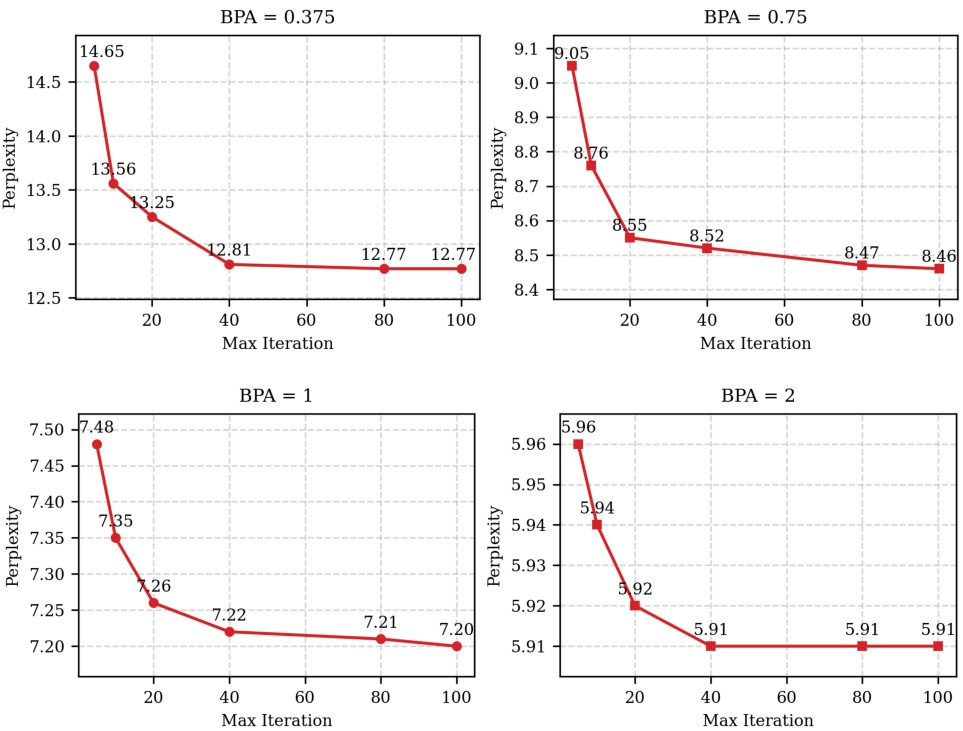

*Figure 8.* **Max iteration sweep.** Empirical convergence of perplexity as a function of the maximum number of iterations. Across all configurations, the algorithm demonstrates efficient convergence, with performance plateauing after 40 iterations.

