# OpenReview forum: "GSRQ: Gain-Shape Residual Quantization for Sub-1-bit KV Cache"
_ICML.cc/2026/Conference — ICML 2026 regular_

### Official Review · Reviewer_TrRY · 2026-02-20

**Soundness:** 3
**Presentation:** 2
**Significance:** 3
**Originality:** 3
**Overall Recommendation:** 4
**Confidence:** 3

**Summary:**

This paper introduces Gain-Shape Residual Quantization (GSRQ), a novel method for compressing the Key-Value cache in Large Language Models to below 1 bit per dimension. It addresses a key limitation of standard K-means used in prior vector quantization methods: in high dimensions, centroid shrinkage weakens directional fidelity, harming reconstruction quality at ultra-low bitrates. The core innovation is Gain-Shape K-means (GSKM), a drop-in replacement that explicitly separates magnitude (gain) and direction (shape) during codebook learning to better preserve angular alignment. Integrated into a residual quantization pipeline with gradient-based weighting, GSRQ significantly outperforms strong baselines, improving average accuracy on LongBench tasks for LLaMA-3-8B from 11.34 to 32.26 at 1 bit.

**Compliance With Llm Reviewing Policy:**

Affirmed.

**Final Justification:**

The author addressed most of my concern, I recommend it for acceptance.

**Key Questions For Authors:**

See weaknesses.

Given the valuable findings presented in this paper and the great experimental results reported, we are very willing to raise our score once the aforementioned concerns are addressed.

**Limitations:**

yes

**Strengths And Weaknesses:**

### Strengths:
1. The paper's core motivation lies in identifying and rigorously analyzing the "centroid shrinkage" phenomenon and its coupling with directional misalignment in standard $l_2$ K-means under high-dimensional settings. This discovery clearly explains why prior VQ/RQ methods struggle at ultra-low bitrates and establishes a solid theoretical foundation for the proposed solution.
1. The proposed Gain-Shape K-means directly addresses the coupled issue of centroid shrinkage and directional misalignment in standard $l_2$ K-means under high-dimensional, low-bitrate settings. This theoretical insight enables robust vector quantization for weakly-structured residuals.
2. ​ GSRQ achieves breakthrough improvements over strong baselines across multiple LLMs and benchmarks. Notably, it raises the average accuracy on LongBench tasks from 11.34 to 32.26 at 1 bit for LLaMA-3-8B, demonstrating its effectiveness in the challenging sub-1-bit regime.

### Weaknesses:
1. My question is whether the identified issue of centroid shrinkage could be mitigated within the standard $l_2$ K-means framework. For instance, by adding a regularization term to penalize small centroid norms or by using a cosine-distance-based objective (spherical k-means)? The authors may provide discussion comparing GSKM against such baseline modifications if they are reasonable.
2. The experimental validation is limited to older model architectures (LLaMA-3-8B, Mistral-7B).​ Testing on newer, state-of-the-art open-source models like Qwen3-8B would be crucial to demonstrate the generalizability and ongoing relevance of the method.
3. Although LongBench is used, synthetic long-context evaluation datasets like RULER are absent. Furthermore, for mathematical reasoning, GSM8K is considered relatively simple; including more challenging datasets like MATH-500, AIME would provide a stricter stress test for the method's impact on complex reasoning.

---

> ### Author Rebuttal · Authors · 2026-03-31
>
> We appreciate the reviewer's insightful suggestions. Below, we address each point.
>
> **1. Can centroid shrinkage be mitigated within standard K-means via regularization or spherical K-means?**
>
> This is an excellent question. To investigate whether the centroid shrinkage issue can be addressed without departing from the standard K-means framework, we tested two regularization strategies (V1, V2) and Spherical K-means in 0.75 BPA.
>
> **Reg V1 (Direct Penalty):**
>
> $\displaystyle\mathcal{L} = \sum_i \lVert x_i - \mu_{k_i} \rVert_2^2 - \lambda \sum_k \lVert \mu_k \rVert_2^2$,
> ($x_i$: $i$-th input, $\mu_{k_i}$: assigned centroid, $\mu_k$: $k$-th centroid)
>
> ||$\lambda$=0.01|$\lambda$=0.05|$\lambda$=0.1|$\lambda$=0.5|$\lambda$=1.0|$\lambda$=2.0|$\lambda$=3.0|
> |:-:|:-:|:-:|:-:|:-:|:-:|:-:|:-:|
> |PPL|13.824|14.102|13.740|14.476|13.919|**13.289**|13.707|
>
> **Reg V2 (Margin-based Penalty):**
>
> $\displaystyle\mathcal{L} = \sum_i \lVert x_i - \mu_{k_i} \rVert_2^2 - \lambda \sum_k \max(0, \tau -  \lVert \mu_k \rVert_2)^2$
>
> To set $\tau$ (margin), we first ran standard K-means and examined the resulting centroid norm distribution, then set $\tau$ separately for each layer and for keys/values accordingly. $\tau_1$ through $\tau_4$ correspond to correcting centroids below the near-zero, 5th, 10th, and 25th percentile of the norm distribution, respectively.
>
> ||$\lambda$=0.01|$\lambda$=0.05|$\lambda$=0.1|$\lambda$=0.5|$\lambda$=1.0|$\lambda$=2.0|
> |:-:|:-:|:-:|:-:|:-:|:-:|:-:|
> |$\tau_1$|12.758|13.314|12.670|12.660|13.144|12.900|
> |$\tau_2$|12.979|12.776|12.612|13.092|13.429|13.403|
> |$\tau_3$|12.878|**12.486**|13.880|13.141|13.790|12.917|
> |$\tau_4$|13.227|13.179|13.215|12.671|14.104|13.460|
>
> We then compared the best configurations of each approach against GSKM:
>
> |Method|MSE|Cosine Similarity|Gain Error|Perplexity|
> |:-|:-:|:-:|:-:|:-:|
> |KM|0.464|0.825|1.702|12.964|
> |Spherical|1.941|0.840|11.937|4437|
> |Spherical + avggain|0.499|0.840|**0.708**|42.674|
> |Reg V1 ($\lambda$=2.0)|0.491|0.825|0.819|13.289|
> |Reg V2 ($\lambda$=0.05, $\tau_3$)|0.464|0.825|1.702|12.486|
> |**GSKM (Ours)**|**0.463**|**0.840**|1.664|**9.327**|
>
> The results suggest that mitigating centroid shrinkage within the standard K-means framework has significant limitations:
> - **Regularization reduces gain error but does not improve PPL.** Reg V1 reduces gain error (1.702 → 0.819), confirming that the regularizer restores centroid scale. However, cosine similarity remains at 0.825 and PPL slightly worsens (12.964 → 13.289), as the modified objective increases MSE (0.464 → 0.491). Reg V2 (12.486) achieves a better PPL than Reg V1 (13.289) but still falls far short of GSKM (9.327).
> - **Spherical K-means improves direction but destroys magnitude.** Spherical K-means matches GSKM in cosine similarity (0.840), confirming that a cosine-based objective recovers better directions. However, it discards magnitude information, leading to catastrophic gain error (11.937) and PPL (4437). Even when magnitude is restored via the average norm of cluster members (Spherical + avggain), PPL remains catastrophic (42.674), indicating that post-hoc magnitude recovery without joint optimization is insufficient.
> - **GSKM achieves the best trade-off.** GSKM is the only method that improves cosine similarity (0.825 → 0.840) while maintaining low MSE (0.463) and gain error (1.664), yielding the best PPL (9.327). This suggests that alternating gain–shape optimization — jointly refining direction and magnitude within each iteration — addresses the problem more fundamentally than penalty terms or cosine similarity alone.
>
> ---
>
> **2. Generalization to newer architectures (Qwen3-8B).**
>
> We thank the reviewer for this suggestion. To test whether the method generalizes beyond the architectures used in the main paper, we additionally evaluated **Qwen3-8B** on **perplexity** using WikiText-2 and C4:
>
> |Method|BPA|WikiText-2|C4|
> |:-:|:-:|:-:|:-:|
> |FP16|16|8.55|8.59|
> |VQLLM|2|9.05|9.00|
> |**GSRQ**|2|**8.88**|**8.80**|
> |VQLLM|1|15.79|13.52|
> |**GSRQ**|1|**10.50**|**9.93**|
>
> GSRQ consistently outperforms VQLLM on Qwen3-8B, confirming that the method generalizes beyond the architectures considered in the main paper.
>
> ---
>
> **3. Evaluation on challenging benchmarks (MATH-500, RULER, AIME)**
>
> We also agree that stronger stress tests are valuable. We therefore additionally evaluated **LLaMA-3-8B-Instruct** on **MATH-500** and **RULER**, and **Qwen3-8B** on **AIME**:
>
> |Method|BPA|MATH-500|RULER|
> |:-:|:-:|:-:|:-:|
> |FP16|16|20.4|96.0|
> |KIVI|2|15.8|94.0|
> |VQLLM|2|13.4|94.0|
> |**GSRQ**|2|**19.0**|**96.0**|
> |KIVI|1|0.8|13.0|
> |VQLLM|1|0.2|20.0|
> |**GSRQ**|1|**14.2**|**72.0**|
>
> |Method|BPA|AIME|
> |:-:|:-:|:-:|
> |-|FP16|36.67|
> |VQLLM|2|16.67|
> |**GSRQ**|2|**33.33**|
>
> These results confirm that GSRQ's advantage extends to harder reasoning and long-context benchmarks beyond the original evaluation set.

---

> > ### Author Rebuttal · Reviewer_TrRY · 2026-04-01
> >
> > The author’s experiments on qwen3-8B and on AIME, MATH500 have made their results more convincing, and we will increase our rating accordingly.

---

> > > ### Author Response · Authors · 2026-04-05
> > >
> > > We sincerely thank the reviewer for the constructive feedback and for raising the score. Your comments have helped us strengthen the paper.

---

### Official Review · Reviewer_yxhd · 2026-02-20

**Soundness:** 3
**Presentation:** 3
**Significance:** 4
**Originality:** 3
**Overall Recommendation:** 4
**Confidence:** 4

**Summary:**

The paper proposes Gain-Shape Residual Quantization (GSRQ) for compressing Key-Value (KV) caches in large language models (LLMs). The method addresses the memory capacity constraints that arise during the decoding phase of long-context generation. Standard Euclidean $K$-means clustering suffers from centroid shrinkage when applied to high-dimensional spaces. To mitigate this issue, the magnitude and direction of the vectors are separated during the codebook update process. A logarithmic weighting scheme is applied to the gradients to stabilize the clustering algorithm against extreme outliers. The quantized cache is decoded on the fly during inference to prevent the model from exceeding GPU memory limits. The empirical results indicate lower perplexity and improved accuracy on various benchmarks compared to existing baselines.

**Compliance With Llm Reviewing Policy:**

Affirmed.

**Final Justification:**

The rebuttal addressed most of my concerns. The discussion on weight quantization methods (W1), calibration size ablation (W2), and the K vs R tradeoff (W4) were all convincingly resolved. However, my question on the theoretical justification for logarithmic smoothing over clipping mechanisms (W3) remains unanswered. The response addressed computational overhead instead, which is a separate question. This gap is not critical enough to change my overall assessment but I encourage the authors to include this comparison in the camera-ready version. I maintain my score of Weak Accept (4).

**Key Questions For Authors:**

1. Could the authors clarify why the literature review omits recent extreme weight quantization methods that also tackle the curse of dimensionality in vector quantization? Incorporating these could strengthen the motivation for choosing a simpler $K$-means adaptation over heavier algorithms for dynamic KV caches.
2. It is unclear if the method remains stable when the calibration dataset size is reduced or increased. Could the authors provide an ablation study on the number of calibration samples to confirm the robustness of the log-weighted smoothing?
3. Could the authors explain if the gain-shape separation introduces any measurable computational overhead during the calibration phase compared to standard Euclidean $K$-means?
4. The end-to-end language model evaluations fix the codebook size at $K=256$. Could the authors provide an ablation study on the codebook size to clarify the optimal trade-off between $K$ and $R$ under a fixed bits-per-activation budget?

**Limitations:**

Yes

**Strengths And Weaknesses:**

**Strengths:**

1. The paper clearly identifies the architectural difference between static weights and dynamic KV caches, justifying the need for a highly efficient quantization scheme rather than complex post-training weight compression algorithms.
2. The mathematical separation of gain and shape effectively prevents centroid shrinkage, which frequently occurs when standard K-means is applied to weakly structured Value vectors in high-dimensional spaces.
3. The method demonstrates strong empirical performance in the extreme sub-1-bit regime, where prior residual vector quantization approaches experience severe performance collapse.
4. The inclusion of hardware-efficient fused kernels confirms that the method avoids the computational bottlenecks typically associated with on-the-fly dequantization.

**Weaknesses:**

1. The introduction does not contextualize the proposed method alongside recent advancements in weight quantization. Discussing methods like lattice-based [1] or trellis-coded [2] quantization could strengthen the motivation for choosing a simpler $K$-means adaptation for dynamic KV caches.
2. The experimental setup relies on 128 calibration samples without providing an ablation study to verify if the model performance is robust to changes in this calibration size.
3. The text lacks a detailed theoretical explanation of why the logarithmic smoothing function bounds the outlier gradients better than alternative clipping mechanisms.
4. The evaluation of the full GSRQ pipeline fixes the codebook size at $K=256$. The paper lacks an end-to-end ablation study on how varying the codebook size impacts the final language model performance under a fixed bit-budget, which leaves the optimal balance between codebook size and the number of residual stages unclear.

**References**

[1] Tseng, Albert, et al. "QuIP# Even Better LLM Quantization with Hadamard Incoherence and Lattice Codebooks." _International Conference on Machine Learning._ 2024.

[2] Tseng, Albert, et al. "QTIP: Quantization with Trellises and Incoherence Processing." _Advances in Neural Information Processing Systems._ 2024.

---

> ### Author Rebuttal · Authors · 2026-03-29
>
> **1. Discussion of extreme weight quantization methods.**
>
> We sincerely thank the reviewer for this insightful suggestion. We will add the following discussion to Section 2 in the revised manuscript:
>
> > **Extreme Weight Quantization via Advanced VQ.** Recent methods such as AQLM (Egiazarian et al., 2024), QuIP# (Tseng et al., 2024), and GPTVQ (van Baalen et al., 2024) push weight quantization into the sub-2-bit regime by combining multi-codebook VQ with heavy offline optimization (e.g., second-order Hessian proxies, incoherence processing, block coordinate descent). These approaches are effective for *static* model weights, where hours-long encoding is amortized over deployment. However, KV cache vectors are generated dynamically at every decoding step, making such solvers impractical — encoding latency would negate any throughput benefit from memory reduction. This runtime constraint motivates our choice of a lightweight K-means adaptation (GSKM) that preserves the speed required for online encoding while addressing centroid shrinkage in high dimensions.
>
> This discussion would help clarify why lightweight K-means adaptations are more suitable than heavy VQ solvers for dynamic KV caches, and better situate GSRQ within the broader VQ landscape.
>
> ---
>
> **2. Robustness of log-weighted smoothing across calibration sizes.**
>
> To verify whether the proposed log-weighted smoothing remains stable under different calibration-set sizes, we evaluated both weighting schemes with varying numbers of calibration samples, $N \in \{16, 32, 64, 128, 256, 512\}$:
>
> |Method|$N$=16|$N$=32|$N$=64|$N$=128|$N$=256|$N$=512|
> |:-:|:-:|:-:|:-:|:-:|:-:|:-:|
> |Raw-Weighted-GSKM|21.4|18.3|16.7|16.0|15.8|16.0|
> |**Log-Weighted-GSKM**|**18.5**|**16.6**|**15.6**|**15.2**|**15.2**|**15.3**|
>
> As shown above, Log-Weighted-GSKM consistently outperforms Raw-Weighted-GSKM across all calibration sizes, with especially clear gains when the calibration set is small. Moreover, its performance improves smoothly and quickly reaches a stable plateau as more calibration samples become available. Overall, these results indicate that the proposed log-weighted smoothing is robust to calibration-set size and does not rely on a narrowly tuned number of calibration samples.
>
> ---
>
> **3. Computational overhead of GSKM.**
>
> GSKM adds two sources of overhead versus standard K-means:
>
> (1) **Per-iteration cost (+9.7 ms):** GSKM performs alternating gain–shape optimization instead of simple arithmetic averaging, adding 9.7 ms per iteration.
>
> (2) **One-time pre/post-processing (+150.4 ms):** L2 normalization (67.1 ms) and gain initialization (83.3 ms) are each executed once per layer, not per iteration.
>
> We profiled per-layer calibration time on LLaMA-3-8B at 0.375 BPA using a single A100 GPU:
>
> |Step|Std K-Means|GSKM|Runs|Complexity|
> |:-:|:-:|:-:|:-:|:-:|
> |L2 Normalization|—|67.1 ms|1|O(N·d)|
> |K-means (assign+update)|79.9 ms|89.6 ms|≤20|O(N·K·d)|
> |Gain Initialization|—|83.3 ms|1|O(N·d)|
> |**Total per layer**|**1,598 ms**|**1,942 ms**|||
>
> The per-layer overhead is modest, with the same asymptotic complexity as standard K-means (Section 5.2). Moreover, the 20-iteration is an upper bound. In most cases, GSKM terminates early when assignments stabilize (Algorithm 1, line 15–16), and in practice most layers converge well before the maximum, further reducing the actual overhead.
>
> Crucially, **calibration is a one-time offline cost** — at inference time, GSRQ uses the identical codebook lookup as any VQ method, adding zero serving latency. This modest cost yields substantial gains (e.g., PPL 21.44 → 15.51 at 0.375 BPA, [Table 6](https://github.com/anonymous117162/Table6)).
>
> ---
> **4. Trade off between K and R under fixed bit budget.**
>
> Under a fixed BPA, we swept the trade off between K (codebook size) and R (number of residual stages) on LLaMA-3-8B with 10 calibration samples (WikiText-2).
>
> |BPA|R|K|D|Codebook size (MB)|Perplexity|
> |:-:|:-:|:-:|:-:|:-:|:-:|
> |0.375|16|64|256|4|19.82|
> |**0.375**|**12**|**256**|**256**|**12**|**15.33**|
> |0.375|8|4096|256|128|15.13|
> |||||||
> |0.750|16|64|128|4|10.08|
> |**0.750**|**12**|**256**|**128**|**12**|**8.53**|
> |0.750|8|4096|128|128|7.90|
> |||||||
> |**1.000**|**16**|**256**|**128**|**16**|**7.14**|
> |0.984|14|512|128|28|6.98|
>
> Increasing K does lower perplexity (e.g., 19.82 → 15.13 at 0.375 BPA), but comes with proportionally larger codebook memory. At K=4096 the codebook grows to 128 MB — a **10× increase** over K=256 (12 MB) — for only marginal PPL improvement (15.33 → 15.13).
>
> We deliberately chose K=256 as a **practical configuration** that balances quality and deployment efficiency. GSRQ with K=256 already substantially outperforms all baselines at matched bit budgets ([Tables 1](https://github.com/anonymous117162/Table1)), demonstrating that our gains stem from the GSKM primitive and RQ pipeline — not from impractically large codebooks.

---

> > ### Author Rebuttal · Reviewer_yxhd · 2026-04-01
> >
> > Thank you for the detailed rebuttal addressing my key questions.
> >
> > I would like to follow up on Weakness 3 regarding the theoretical explanation of logarithmic smoothing.
> >
> > Specifically:
> > 1. What are the advantages of log transformation compared to simple clipping?
> > 2. Can the authors provide a mathematical explanation of how log transformation handles heavy-tailed gradient distributions more effectively?
> > 3. Does log-weighting mitigate the threshold selection problem that clipping faces?
> >
> > I would appreciate clarification on these points.

---

> > > ### Author Response · Authors · 2026-04-05
> > >
> > > We appreciate the reviewer's request for further clarification and are pleased to provide a more detailed explanation below.
> > >
> > > **1. Advantages of log transformation over simple clipping**
> > >
> > > The key advantage of log transformation is that it suppresses outliers **without collapsing the importance ordering among salient tokens**. Clipping applies a hard truncation $\tilde{w}_i = \min(w_i, c)$, mapping all values above the threshold to the same cap and thus removing relative ordering among highly important samples. Our log smoothing instead applies a smooth, monotone compression that preserves a softer ranking among salient tokens while still suppressing the heavy tail—better suited to low-bit KV cache quantization where codebook misallocation is costly.
> > >
> > > **2. Mathematical explanation**
> > >
> > > In weighted GSKM, per-sample weights are gradient norms $w_i = \|\nabla_{x_i}\mathcal{L}\|_2$, which are highly heavy-tailed (max/median > 400×, [Figure 3](https://github.com/anonymous117162/Figure3)). Since these weights enter linearly in the shape update, raw weighting allows a few outliers to dominate the learned codewords; empirically, Raw-Weighted-GSKM (PPL=16.08) even *underperforms* Unweighted-GSKM (PPL=15.51) in [Table 5](https://github.com/anonymous117162/Table5).
> > >
> > > This is where log smoothing differs from clipping. For $w_a \gg w_b$, raw weighting yields an influence ratio proportional to $w_a/w_b$, which can be arbitrarily large. Clipping uses $f_{\mathrm{clip}}(w)=\min(w,c)$, so once both exceed $c$, their transformed weights become identical and the ratio collapses to 1, discarding the ordering among large weights. By contrast, our transformation $\tilde{w}_i=\log(1+\lambda w_i)$ with $\lambda=\tau/(\mathrm{median}(w)+\epsilon)$ remains strictly increasing while smoothly compressing large values.
> > >
> > > The effectiveness of this function stems from two properties:
> > >
> > > > **Property 1: Tail Compression.** Since $f''(w) = -\lambda^2/(1+\lambda w)^2 < 0$, the function is strictly concave. For large $\lambda w \gg 1$, it behaves as $\log(1+\lambda w) \approx \log(\lambda w)$, so a multiplicative gap of $\alpha$ in raw weights compresses to an additive $\log \alpha$ contribution. This prevents extreme-gradient samples from monopolizing centroid placement without flattening them to a single cap.
> > >
> > > > **Property 2: Order Preservation.** Since $f'(w) = \lambda/(1+\lambda w) > 0$ for all $w \geq 0$, the importance ordering is unconditionally preserved: $w_a > w_b \Rightarrow \tilde{w}_a > \tilde{w}_b$. This guarantees that more loss-sensitive tokens always exert greater influence in the weighted shape update.
> > >
> > > Together, these properties make log smoothing a better fit for heavy-tailed gradient distributions: it reduces outlier dominance more gently than raw weighting, while avoiding the ranking collapse introduced by clipping.
> > >
> > > **3. Log-weighting mitigates the threshold selection problem**
> > >
> > > A practical challenge of clipping is that a single threshold $c$ may not generalize well across the diverse gradient distributions encountered at different layers and residual stages. To validate how sensitively performance depends on this choice, we swept the clipping threshold $c$ and report the resulting perplexity:
> > >
> > > | | $c=10^{-6}$ |$c=10^{-5}$|$c=10^{-4}$|$c=10^{-3}$|$c=10^{-2}$|
> > > |-|-|-|-|-|-|
> > > |BPA=0.375|15.63|**15.43**|15.53|16.17|16.25|
> > > |BPA=0.75|9.15|9.07|**8.93**|8.95|9.01|
> > > |BPA=1.0|7.49|7.44|7.39|**7.37**|7.38|
> > >
> > > The optimal threshold is **not consistent across bit rates** ($c=10^{-5}$ is best at BPA=0.375; $c=10^{-4}$ at BPA=0.75 and $c=10^{-3}$ at BPA=1.0), confirming that clipping requires per-setting tuning. By contrast, our method matches or slightly improves the best clipping baseline, with clearer gains at lower BPA. This is consistent with the smooth concave form of log smoothing: unlike clipping, it does not introduce a hard boundary at which all larger weights are collapsed to the same value.
> > >
> > > We additionally swept $\tau$ in our method. (A more detailed analysis of $\tau$ is provided in [Appendix B](https://github.com/anonymous117162/AppendixB)).
> > >
> > > | |$\tau=0.1$|$\tau=0.5$|$\tau=0.7$|$\tau=1.0$|$\tau=3.0$|$\tau=10.0$|
> > > |-|-|-|-|-|-|-|
> > > |BPA=0.375|15.44|15.38|15.31|**15.20**|15.31|15.50|
> > > |BPA=0.75|8.86|8.87|8.86|**8.85**|8.91|8.95|
> > > |BPA=1.0|7.37|7.37|7.36|**7.36**|7.39|7.39|
> > >
> > > Unlike clipping—where performance is more sensitive to the threshold choice—deviations from $\tau=1.0$ yield only marginal changes, suggesting that while our method still requires choosing $\tau$, log smoothing mitigates the threshold selection problem and delivers consistently better perplexity.

---

### Official Review · Reviewer_a3wb · 2026-02-21

**Soundness:** 2
**Presentation:** 2
**Significance:** 2
**Originality:** 3
**Overall Recommendation:** 4
**Confidence:** 4

**Summary:**

This work proposes an approach for vector quantization of KV-cache, where codeword centroids are represented as a composition of magnitude - gain and angle - shape. This parametrization is claimed to mitigate the centroid shrinkage problem as well as the ignorance of angle in high dimensions. In addition, authors propose logarithmically transformed gradient weighting for sensitivity weighting. The introduced approach is validated on a key-value compression for several LM models (Llama-⅔, Mistral) at extreme compression ratios (0.375-2 bit per value).

**Compliance With Llm Reviewing Policy:**

Affirmed.

**Final Justification:**

After reading the responses addressed to me and other reviewers I am leaning towards acceptance.
The new results provide additional evidence for the applicability and practicality of the method.

**Key Questions For Authors:**

- The approach uses large subspace dimension and large number of residuals. How does the method perform for smaller vector sizes (say 8 or 16)?

**Limitations:**

-

**Strengths And Weaknesses:**

Strengths

- The insights about the centroid shrinkage / angular sensitivity appear to be novel and overlooked in prior art on vector quantization.
- The proposed approach achieves pretty decent performance, especially at high compression ratio. The perplexity increases only twofold for 0.375 bit per parameter.
- The implementation is accompanied by dedicated triton kernels that are claimed to provide speed-ups.

Weaknesses

- Some important baselines are missing. As a baseline, [1] considers vector quantization with HIGGS [2] grids (optimal for Gaussian distributed weights). I would recommend conducting a comparison with this method at 2 bit compression. In addition, it would be insightful to know whether codewords for HIGGS grid suffer the same problem.
- There are no details about the inference framework for speed-up measurement. Is it transformers + pytorch code? What are the end-to-end speedups in terms of decoding latency for different sequence lengths?
- The code is not provided in the supplementary, making one unable to verify the claims.

References

---
[1] Shutova, Alina, et al. "Cache me if you must: Adaptive key-value quantization for large language models." arXiv preprint arXiv:2501.19392 (2025).

[2] Malinovskii, Vladimir, et al. "Pushing the limits of large language model quantization via the linearity theorem." arXiv preprint arXiv:2411.17525 (2024).

---

> ### Author Rebuttal · Authors · 2026-03-31
>
> **1. Comparison with HIGGS grids.**
>
> Thank you for the recommendation. HIGGS targets weight quantization, so we reproduced its method for KV cache quantization and conducted a direct comparison on LLaMA-3-8B (WikiText-2):
>
> |Bit|HIGGS Perplexity|GSRQ Perplexity|
> |:-:|:-:|:-:|
> |FP16|5.54|**5.54**|
> |4|5.62|**5.59**|
> |3|5.91|**5.73**|
> |2|8.31|**5.93**|
> |1|292.02|**7.36**|
>
> **Performance comparison.** HIGGS achieves competitive perplexity in a completely data-free manner, which speaks to its strong design. Nonetheless, GSRQ consistently outperforms HIGGS across all bit rates. At 2-bit — the setting the reviewer specifically suggested — GSRQ reduces PPL from 8.31 to 5.93, a substantial gap. We attribute this to: (1) HIGGS uses **static, data-independent grids**, so some codewords may go underutilized for a given input's KV distribution, wasting representational capacity; (2) without calibration data, HIGGS **cannot adapt to the actual activation distribution**, whereas GSRQ's learned codebooks are tailored to the target model's statistics.
>
> **Centroid shrinkage analysis for HIGGS.** HIGGS does not suffer from centroid shrinkage because it uses mathematically pre-defined lattice grids rather than learned codebooks — there is no arithmetic averaging step that could induce cancellation in high dimensions. This corroborates our core thesis: centroid shrinkage is a limitation of ℓ₂-based K-means centroid updates, and GSKM's gain–shape decomposition is designed to address it.
>
> ---
>
> **2. Inference Framework and End-to-End Latency.**
>
> We thank the reviewer for this important suggestion. We acknowledge that the paper should make the inference framework more explicit.
>
> Our measurements are obtained from a **fully custom PyTorch-based autoregressive decoding implementation** with **custom Triton kernels**. In particular, we use fused kernels for **(i)** residual quantization + bit-packing of newly generated KV vectors, and **(ii)** on-the-fly bit-unpacking, dequantization, RoPE, and attention computation during decoding. For long contexts, we additionally use a *Split-KV (FlashDecoding-style)* parallelization strategy.
>
> To directly answer the reviewer’s question, we additionally measured the full autoregressive decoding latency on **LLaMA-3-8B** under BPA=0.75, using *torch.cuda.Event* timers with synchronization.
>
> **End-to-end decoding latency per token**
> *(single A100-80GB GPU, batch size=1, generated tokens=100)*
>
> |Context length|FP16 (ms)|GSRQ (ms)|Speedup|
> |:-:|:-:|:-:|:-:|
> |8K|70.68|44.48|**1.59x**|
> |16K|122.50|54.47|**2.25x**|
> |32K|253.69|83.96|**3.02x**|
> |64K|486.73|143.29|**3.40x**|
> |128K|**OOM**|262.12|--|
>
> Overall, we observe substantial latency reductions at long contexts. Compared to FP16, GSRQ achieves a **1.59x-3.40x speedup** in per-token decoding latency over the evaluated range, while FP16 runs out of memory at 128K.
>
> ---
>
> **3. Code Availability.**
>
> We agree that code availability is important for verifying empirical claims. We have made the implementation available at:
>
> - **Code:** https://github.com/anonymous117162/GSRQ.git
>
> This includes the **GSKM** implementation and the evaluation pipeline for comparing reconstruction quality under **MSE**, **Gain Error**, and **Cosine Similarity**. The reviewer can directly inspect the code and verify the comparisons reported in the paper. The full GSRQ pipeline and evaluation scripts will be released upon acceptance.
>
> ---
>
> **4. Performance at small subspace dimensions.**
>
> We evaluated GSRQ with small subspace dimensions (D=8, 16) on LLaMA-3-8B (WikiText-2):
>
> |BPA|R|K|D|PPL (K-means)|PPL (GSRQ)|
> |:-:|:-:|:-:|:-:|:-:|:-:|
> |FP16|-|-|-|5.54|5.54|
> |4|4|256|8|5.79|5.79|
> |4|8|256|16|5.69|5.67|
> |2|2|256|8|6.72|6.71|
> |2|4|256|16|6.52|6.51|
> |1|1|256|8|14.43|14.49|
> |1|2|256|16|11.51|11.51|
>
> Standard K-means and GSRQ achieve nearly identical perplexity at small D, which supports our core thesis. Centroid shrinkage is fundamentally a high-dimensional effect — at D=8 or 16, within-cluster vectors remain well-aligned, so averaging cancellation is minimal. This is directly observable in [Figure 5](https://github.com/anonymous117162/Figure5) : cosine similarity is close to 1.0 for both methods at small D.
>
> The benefit of GSKM emerges when D is large enough for angular dispersion to cause meaningful shrinkage — precisely the regime required for sub-1-bit rates, where large D is necessary to keep residual stages manageable (e.g., D=128 for 1-bit, D=256 for 0.375-bit). Most prior VQ-based KV cache methods operate with small subspace dimensions where this problem does not manifest; pushing toward sub-1-bit rates exposes the failure mode, and GSKM provides an effective remedy.

---

> > ### Author Rebuttal · Reviewer_a3wb · 2026-04-01
> >
> > Thank you for the response.
> >
> > The concerns were addressed, therefore I decide to raise my score.

---

> > > ### Author Response · Authors · 2026-04-05
> > >
> > > We thank the reviewer for the helpful feedback and for updating the score. Your comments have made our paper stronger.

---

### Official Review · Reviewer_W8Aq · 2026-03-11

**Soundness:** 3
**Presentation:** 2
**Significance:** 3
**Originality:** 3
**Overall Recommendation:** 3
**Confidence:** 4

**Summary:**

This work reveals a problem with k-means clustering used in traditional VQ methods, i.e., centroid shrinkage, and propose a novel Gain-Shape K-means clustering method to address the problem. I think the improved K-means clustering method is novel, and numerical experiments demonstrate its effectiveness.

**Compliance With Llm Reviewing Policy:**

Affirmed.

**Key Questions For Authors:**

See weakness

**Limitations:**

yes

**Strengths And Weaknesses:**

### Strengths
1.The improved K-means clustering method is novel and effective.

### Weaknesses
1.There is a lack of explanation for motivation. The centroid shrinkage problem seems to be a matter of scale, and has little to do with the angle mentioned in the first paragraph of Section 4.
2.The paper does not provide a perspective for understanding the proposed K-means clustering method, such as the definition of centroid (3), and the shape (7) and gain update (8).
3.Why does the proposed K-means clustering method (Algorithm 1) converge?

---

> ### Author Rebuttal · Authors · 2026-03-31
>
> **1. Clarification on the relationship between centroid shrinkage and angular alignment.**
>
> We thank the reviewer for this insightful comment. We agree that centroid shrinkage is most directly a **scale** phenomenon, and that the connection to angular degradation deserves a more careful explanation. Our intention was not to claim that this dynamics has been fully established, but rather to present it as a **hypothesis** for why standard K-means becomes particularly fragile in high-dimensional settings.
> To better understand whether the issue is primarily about scale or direction, we conducted additional experiments using a regularized K-means variant that explicitly counteracts scale loss:
>
> $\displaystyle\mathcal{L} = \sum_i \lVert x_i - \mu_{k_i} \rVert_2^2 - \lambda \sum_k \lVert \mu_k \rVert_2^2$,
> ($x_i$: $i$-th input, $\mu_{k_i}$: assigned centroid, $\mu_k$: $k$-th centroid)
>
> which admits a closed-form centroid update and directly restores centroid norms. If the problem were purely one of scale, this should recover most of the performance. We swept $\lambda$ and report the best-performing configuration ($\lambda$=2.0; full sweep in our response to Reviewer TrRY, Q1) below (0.75 BPA, LLaMA-3-8B, WikiText-2):
>
> |Method|MSE|Cosine Similarity|Gain Error|Perplexity|
> |:-|:-:|:-:|:-:|:-:|
> |KM|0.464|0.825|1.702|12.964|
> |Regularized KM|0.491|0.825|**0.819**|13.289|
> |**GSKM (ours)**|**0.463**|**0.840**|1.664|**9.327**|
>
> *Regularized K-means* successfully reduces the gain error (1.702 → 0.819), confirming that the regularizer restores centroid scale as intended. However, its cosine similarity remains identical to standard KM (0.825), and the downstream PPL even worsens (12.964 → 13.289). In contrast, GSKM improves cosine similarity (0.825 → 0.840) with a comparable gain error, and this directional improvement translates into a large PPL reduction (12.964 → 9.327).
>
> These results suggest that correcting scale alone is insufficient — effectively addressing centroid shrinkage requires considering both scale and angular alignment, which is precisely the motivation for our gain–shape formulation.
>
> ---
>
> **2. Additional perspective on centroid parameterization, shape update, and gain update.**
>
> We agree with the reviewer that Section 5.2 may not have provided enough intuition for the proposed gain–shape decomposition, including why we parameterize the centroid as in (3), how the shape and gain updates in (7)–(8) are obtained, and what objective the shape update is intended to optimize. We therefore provide a more detailed clarification below.
>
> - **Centroid parameterization:**
> Each centroid is written as $\mu_k = g_k s_k$ ($g_k \geq 0$, $\lVert s_k \rVert_2 = 1$). This represents the same centroid, but separates direction and magnitude so that each can be updated with a tailored rule. In particular, it enables a magnitude-invariant direction estimate that standard K-means does not provide.
>
> - **Shape update:**
> We solve $ \max_{\lVert s \rVert_2=1} \sum_{i \in I_k} \cos(x_i, s).$
> Letting $y_i = x_i / \lVert x_i \rVert_2$, this becomes $\max_{\lVert s \rVert_2=1} \sum_i y_i^\top s$,
> whose closed-form solution is $s_k \propto \sum_i y_i$ by *Cauchy–Schwarz*. Normalizing each vector before averaging removes the influence of individual magnitudes, thereby decoupling direction estimation from shrinkage.
>
> - **Gain update:**
> With $s_k$ fixed, minimizing $\sum_{i \in I_k} \lVert x_i - g_k s_k \rVert_2^2$ with respect to $g_k$ gives $g_k = \frac{1}{|I_k|}\sum_{i \in I_k} x_i^\top s_k$,
> i.e., the mean projection onto $s_k$. The $\max(0,\cdot)$ clamp enforces $g_k \geq 0$.
>
> ---
>
> **3. Convergence of GSKM ([Algorithm 1](https://github.com/anonymous117162/Algorithm1)).**
>
> This is a very important question, and we agree that a formal convergence guarantee would strengthen the paper. Because the shape step optimizes an angular surrogate while the gain step follows the $\ell_2$ structure, GSKM is not a strict block coordinate descent method for a single objective, and we therefore do not claim monotonic decrease.
>
> To examine this more carefully, we tested two theoretically safer alternatives: (a) GSKM with a descent safeguard that accepts a shape update only when the $\ell_2$ distortion does not increase, and (b) the regularized variant from Q1, which inherits Lloyd-style convergence guarantees. Both perform noticeably worse than GSKM:
>
> |Method|Convergence guarantee|Perplexity|
> |:-|:-:|:-:|
> |Descent safeguard|✓ (monotonic decrease)|12.899|
> |Regularized|✓ (Lloyd-convergent)|13.289|
> |**GSKM (ours)**|—|**9.327**|
>
> This suggests that GSKM benefits from allowing directional corrections that stricter updates suppress. At the same time, it is stable in practice: Appendix D [Table 8](https://github.com/anonymous117162/Table8) shows nearly identical performance at 20 and 40 iterations (15.51 vs. 15.52). Our claim is empirical effectiveness and practical stability, while formal convergence analysis remains important future work.

---

### Decision · Program_Chairs · 2026-04-30

**Decision:**

Accept (regular)

**Comment:**

This submission proposes a gain-shape residual quantization approach for sub-1-bit KV-cache compression. The reviewers recognized the novelty of the centroid shrinkage analysis and the strong empirical performance of the method, particularly in extreme compression regimes, while raising some concerns regarding theoretical justification, missing baselines, and the breadth of the evaluation. The authors provided a detailed response that addressed most of these concerns. Therefore, I recommend acceptance.